# Theory of branching morphogenesis by local interactions and global guidance

Mehmet Can Uçar [1,7 ✉], Dmitrii Kamenev[2,7], Kazunori Sunadome[3], Dominik Fachet [1,6], Francois Lallemend [2,4], Igor Adameyko [3,5], Saida Hadjab [2,8 ✉] & Edouard Hannezo [1,8 ✉]

Branching morphogenesis governs the formation of many organs such as lung, kidney, and the neurovascular system. Many studies have explored system-specific molecular and cellular regulatory mechanisms, as well as self-organizing rules underlying branching morphogenesis. However, in addition to local cues, branched tissue growth can also be influenced by global guidance. Here, we develop a theoretical framework for a stochastic self-organized branching process in the presence of external cues. Combining analytical theory with numerical simulations, we predict differential signatures of global vs. local regulatory mechanisms on the branching pattern, such as angle distributions, domain size, and space-filling efficiency. We find that branch alignment follows a generic scaling law determined by the strength of global guidance, while local interactions influence the tissue density but not its overall territory. Finally, using zebrafish innervation as a model system, we test these key features of the model experimentally. Our work thus provides quantitative predictions to disentangle the role of different types of cues in shaping branched structures across scales.

[1] Institute of Science and Technology Austria, Am Campus 1, 3400 Klosterneuburg, Austria. [2] Department of Neuroscience, Karolinska Institutet, 17177 Stockholm, Sweden. [3] Department of Physiology and Pharmacology, Karolinska Institutet, 17177 Stockholm, Sweden. [4] Ming-Wai Lau Centre for Reparative Medicine, Stockholm node, Karolinska Institutet, Stockholm, Sweden. [5] Department of Neuroimmunology, Center for Brain Research, Medical University of Vienna, 1090 Vienna, Austria. [6] Present address: IRI Life Sciences, Humboldt-Universität zu Berlin, 10115 Berlin, Germany. [7] These authors contributed equally: Mehmet Can Uçar, Dmitrii Kamenev. [8] These authors jointly supervised this work: Saida Hadjab, Edouard Hannezo. ✉email: mehmetcan.ucar@ist.ac.at; saida.hadjab@ki.se; edouard.hannezo@ist.ac.at

Branching morphogenesis is a ubiquitous developmental process, where a number of morphogenetic events cooperate to give rise to complex tree-like morphologies. Branched structures are observed both at the level of multicellular organs, such as lung, kidney, mammary gland or vascular system[1–5], and at the level of single cells such as neurons[6] or tracheal cells[7]. A number of studies in the past decades have been devoted to understanding their design principles, with a particular emphasis on how given branched topologies and geometries can optimize properties such as transport and robustness[8–17].

A complementary question has been to understand the dynamical mechanisms through which branching complexity can arise during development. It has been shown in particular that branching morphogenesis proceeds via tip-driven growth and/or side branching events, which are controlled by combinations of deterministic and stochastic rules[4,18–20]. Indeed, different cellular strategies have been demonstrated to regulate the final branching pattern, from stereotypic transcription factor expression[21], stochastic local rules[22,23], mechanical forces and local reaction-diffusion mechanisms[2,3,24] to epigenetic mechanisms[25] and codes of cell adhesion molecules[19,26]. In addition to these intrinsic mechanisms, branching morphogenesis is also controlled externally by a number of guidance cues from the environment[27–30], including chemical gradients (chemotaxis from diffusible factors or haptotaxis from substrate-bound adhesion or guidance molecules) or gradients in the mechanical stiffness of the environment[31,32]. However, a theoretical framework to quantitatively assess the contribution of each intrinsic and/or extrinsic cue in shape, orientation and size of branched structures, as well as the relative roles of deterministic vs. stochastic factors during branching morphogenesis remains to be established.

Here, we combine numerical simulations with analytical theory to derive a comprehensive description of branching morphogenesis in the presence of internal self-organizing cues (such as self-avoidance of branches, stochastic exploration of space, and tip termination) and external guidance cues. Furthermore, we identify several metrics, including branch directionality, shape or efficiency of space filling, which are differentially affected by different model parameters. These metrics thus provide generic criteria, measurable from static data on the final branched structure, to distinguish different dynamical mechanisms at play during morphogenesis. Finally, we experimentally test our model in peripheral sensory system focusing on the branching of individual Rohon-Beard sensory neurons in the zebrafish caudal fin. Thus, we present a model where the combination of two simple parameters, for local self-interactions and global guidance, can synergize to generate complex branched structures both in two and three-dimensions.

## Results

**Influence of self-avoidance, stochasticity and external guidance on the morphology of branched networks.** To analyze the influence of both the local self-organizing (intrinsic) cues and the global (extrinsic) guidance on the formation of branched structures, we first turned to a modelling approach inspired by the physics of branching random walks, which represents tips as particles undergoing both stochastic and deterministic elongation movements (which generates branches at a constant speed), as well as stochastic branching events into two tips with probability $p_b$. This type of model[20,22,33–35] has the advantage of coarsening many microscopic features of branching regulation (for instance that have been addressed via reaction-diffusion models[36,37]) into simple sets of rules. In this work, we include both the possibility for global guidance via gradients quantified by a guidance strength $f_c$ (which acts as an external force on tip motion) as well as local self-

avoidance of neighboring branch segments. Such self-avoidance can typically occur in neurons by cycles of contact-retraction when a tip touches a neighboring branch of the same cell[20,38], or in branched multicellular organs via diffusible molecules[39]. Here we concentrate on the morphogenesis of single neurons, and therefore model self-avoidance effectively by tips moving deterministically away from neighboring branches of the same tree at strength $f_s$. If the tip fails to reorient in close proximity to a neighboring branch, it terminates its growth and becomes irreversibly inactive (which we call termination/annihilation, see Fig. 1a,b for a schematic of the model). In particular, the effect of the external field on the branch tip can be described by a force acting on the alignment angle $\psi$ between the tip polarity and the field. Microscopically, the external field applies a torque on the tip leading to its reorientation determined by a factor $-f_c \sin(\psi)$, reminiscent of models for dry active particles with polarity interactions[40] and colloidal flocks in an external flow field[41]. Accordingly, we implement in the simulations external guidance as modifying the transition probabilities on the direction of elongation, to bias growth in the direction of the external field: "forward" and "backward" jumps that respectively increase and decrease the alignment angle $\psi$ are determined by the probabilities $p_e A(\psi)$ and $p_e B(\psi)$, where $p_e \equiv 1 - p_b$ is the elongation probability and the reweighting factors $A(\psi)$ and $B(\psi)$ are determined by $A(\psi) - B(\psi) = -f_c \sin(\psi)$ and $A(\psi) + B(\psi) = 1$. Furthermore, we implement self-avoidance by deterministic displacements of the active tip at position $\mathbf{r}$ by a "self-interaction" force $-f_s \mathbf{p_s} \propto -f_s \sum_j (\mathbf{r} - \mathbf{r}_j)$ with $f_s < 0$, pointing away from the density gradient of neighboring branches (at positions $\mathbf{r}_j$ within a radius $R_s$ i.e., $|\mathbf{r} - \mathbf{r}_j| < R_s$), see Supplementary Note 2 for details.

To determine how the morphology and shape of a branching structure is affected by local intrinsic vs. global extrinsic cues, we then asked whether the two key parameters of local self-avoidance $f_s$ and of external guidance $f_c$ could give rise to qualitatively different types of morphologies. Indeed, building a phase diagram of branching morphologies revealed key differences: in the presence of an external, axially oriented (linear) gradient, branched structures adopt triangular shapes, branching in a cone-shape with a well-defined angle that becomes smaller for increasing guidance strength $f_c$ (Fig. 1c). On the other hand, changing the self-avoidance strength $f_s$ gave rise to denser branches with increased local alignment, but did not markedly change the overall shape.

**Derivation of the continuum model.** To back the qualitative insights of the morphology diagram obtained from simulations more quantitatively, we sought to develop an analytical theory of branching via external guidance, which falls under the class of branching and interacting random walks[22] in an external field. Starting from a microscopic description of branching and elongation events, we derived a (continuum) Fokker–Planck equation for the tip growth and branching under the influence of external field guiding elongation (as described in detail in Supplementary Note 1, see also Supplementary Figs. 1–3). In particular, we obtained an equation for the time evolution of the probability of a tip to grow in a given direction, as determined by the alignment angle $\psi$ relative to the polarity of the external field. In the absence of an external field, this direction is subjected to two types of random fluctuations: a branch tip undergoing elongation exhibits a small rotational diffusion that is described by small continuous changes in the alignment angle (bounded by $\psi_e$ at each time point), while branching events lead to abrupt jumps of larger maximal magnitude $\psi_b$ in the branch orientation. To account for the latter, non-local changes in the alignment angle $\psi$, we turned to the theory of Lévy flights, where a generalized Fokker–Planck equation has been proposed[42]. Crucially, using a jump size

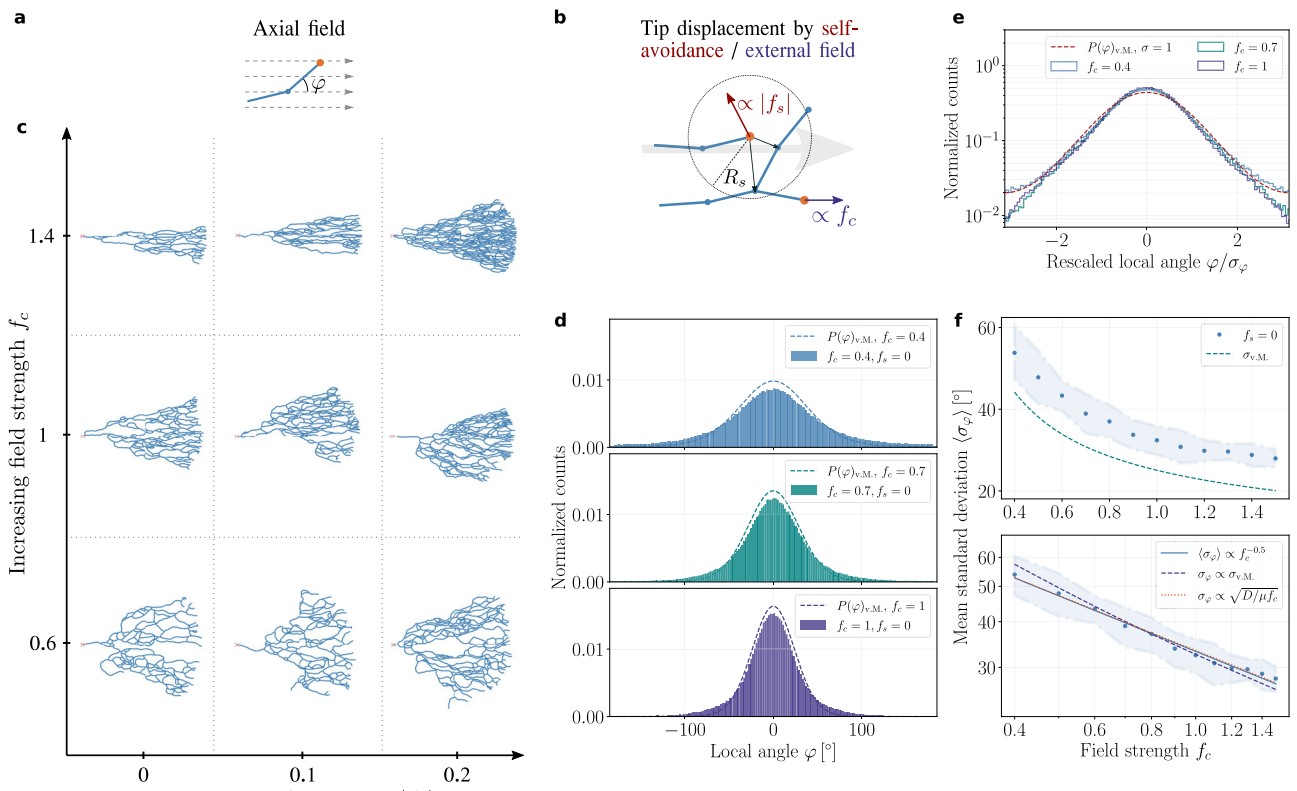

**Fig. 1 Morphology and alignment of branching structures in the presence of global guidance cues and local self-repulsion. a–c** Schematic of the model and resulting branching morphologies. **a** We consider an active tip (orange node) which undergoes stochastic branching and elongation according to a local angle $\varphi$ to make branch segments (blue solid lines), guided by an external field (linear guidance, dashed arrows). **b** Self-avoidance or external field are implemented in the simulation by additional displacements of active tips of the branching network respectively by "sensing" neighboring branch segments (blue nodes) within a radius of repulsion $R_s$ (red arrow in **b**), or by a bias toward the external field (large gray arrow in **b**). The strength of local self-avoidance and external guidance are respectively determined by a factor $|f_s|$ or $f_c$. **c** Morphology diagram of branching and annihilating random walks (BARWs) with linear (axial in one-dimension) external guidance obtained from simulations. Representative networks are displayed for different values of the external field strength $f_c$ and self-avoidance $|f_s|$. **d** Probability distribution of tips growing with an angle $\varphi$ for different values of $f_c$ and without self-repulsion ($f_s = 0$) in the simulations (solid bars). These are well-approximated by the analytical predictions (dashed lines) following a von Mises distribution centered around zero and with single parameter $\nu \equiv \frac{\mu f_c}{D}$ (with $D \simeq 0.03$ and $\mu \simeq 0.18$ as predicted from theory). With increasing field strength $f_c$ the distributions become sharper, indicating better alignment of the branch segments with the external field. **e** Histograms of the local angle displayed in (**d**) rescaled by their corresponding standard deviations (SDs), showing that they all collapse onto the von Mises distribution with unit SD (dashed line), as predicted analytically. **f** Fluctuations in local angle $\sigma_\varphi$ decrease monotonically with increasing field strength $f_c$ (top panel), and are consistent with a power-law relation (bottom panel), close to the scaling law predicted by the analytical theory (dashed line) and to the scaling law $\sigma_\varphi \propto \sqrt{D/(\mu f_c)}$ (dotted line).

distribution $\lambda(\psi - \psi')$ to describe the difference in the alignment angle before $\psi'$ and after $\psi$ an elongation or branching event, we could integrate these two sources of stochasticity into macroscopic "diffusion" and "mobility" coefficients $D \equiv \frac{1}{6}(p_b(\psi_b^2 + \psi_b \psi_e) + \psi_e^2)$ and $\mu \equiv \frac{1}{2}(p_b \psi_b + \psi_e)$, respectively. Finally, consistent with the implementation of the simulations, the effect of the external guidance could then be incorporated by a drift term $-f_c \sin(\psi)$, leading to the Fokker–Planck equation:

$$\partial_t P(\psi, t) = D\partial_\psi^2 P(\psi, t) + \mu \partial_\psi[P(\psi, t)f_c \sin(\psi)], \quad (1)$$

which reflects a sinusoidal reorientation of the active tip by the external field[43].

**Comparison between analytical model and simulations.** Importantly, Eq. (1) describing the probability of branch orientation attains a steady-state solution ($\partial_t P^{st}(\psi, t) = 0$) that is largely independent of the form of the external field, i.e., it applies generically to different geometries after defining the relative branch alignment with respect to the external field. This solution predicts that the alignment of angles with respect to the polarity of the external field will be determined by the von Mises

distribution (circular normal distribution[44]):

$$P(\psi)_{v.M.} = \frac{1}{2\pi I_0(\nu)} \exp(\nu \cos(\psi)), \quad (2)$$

with a concentration parameter given by $\nu \equiv \mu f_c/D$, and $I_0(\nu)$ is the modified Bessel function of the first kind of order zero. The fluctuations in the angular alignment as determined by the variance will thus follow a universal scaling approximately given by $\sigma^2 \propto D/\mu f_c$ that underlines the relative contribution of the local noise to the external guidance. For an axial (linear) field parallel to the horizontal axis, for instance, the above solution applies to the distribution of the local angles $\varphi$ of the branch segments (Fig. 1d). In a radial external field emerging from a central point of origin, however, the alignment of a branch is determined by the angle difference $\psi \equiv \varphi - \theta$ between its local angle $\varphi$ and its angle $\theta$ with respect to the origin of the external field, and thus $\psi$, rather than $\varphi$, is predicted to follow the von Mises distribution (see Supplementary Figs. 4–9 for the alignment angles for different model assumptions). Comparing these analytical criteria with the numerical simulations led to excellent agreement without using any fit parameter (Fig. 1d–f).

To summarize, the combination of analytical modelling and numerical simulations allows us to make a number of predictions for different topological and geometrical properties of branched structures: A key signature of external guidance is that branching angles should conform to a von Mises distribution while the overall branching structure in an axial field can adopt a well-defined conical domain in the absence of defined boundaries. Furthermore, a signature of the stochastic nature of branching and annihilating random walks is that even with external guidance, the local branch length or overall network size and shape should be highly variable to random branching events and local density-driven termination of tip growth.

**Sensory neuron morphogenesis as a biased and branching random walk.** To sequentially test these theoretical predictions, we examined the morphology of sensory neurons using zebrafish caudal fin innervation as a model system, as it has several advantages: (i) it is a simple quasi two-dimensional (see Supplementary Movie 1) and transparent system, facilitating imaging and reconstruction, (ii) the innervation pattern is complex, with tens to hundreds of branches per neuron, and (iii) multiple axons arise from dorsal part of spinal cord and start branching out in a simple geometry, i.e., a roughly semi-circular region (Fig. 2a). To segment and reconstruct single branched neurons, we used genetic sparse labelling strategy to label individual neurons (mCherry positive, see Supplementary Fig. 10) at 5 days post-fertilization (a time when neurons are functional and the fish is able to swim), and skeletonized the manually traced filaments to generate hierarchical tree topologies (see Methods below and Supplementary Note 3 for details). Interestingly, we found that these neurons, although all appearing to grow radially toward the outer edge of the fin, were highly stochastic and heterogeneous both in shape, size, and morphology (Fig. 2c, Supplementary Fig. 11). This hints at a highly stochastic pattern of fin innervation, as expected in our branching random walk model when we adapted it to a radial external field (Fig. 2b). Qualitative comparisons with different stochastic simulations with identical model parameters revealed similar stochasticity in shape, angles, topology and size of neuronal trees, as seen in the experimental data (Fig. 2c–d). Furthermore, few crossovers between branches could be observed with terminal tips residing all over the neuronal structure close to neighboring branches (Fig. 2a,c), as qualitatively expected in the framework of branching and annihilating random walks. Finally, and more quantitatively, we extracted (i) the branch length distribution across neurons, and found that it was very wide (with branches of all lengths seen in data) and well-described by a simple exponential, as predicted by a stochastic branching process (Fig. 2e), and (ii) the size distribution of subtrees (defined as the number of branches derived from a given branch point, looked at for all branch points at any generation number) showing similar and long-tailed distributions in both data and simulations (Supplementary Fig. 12). Altogether these key features supported the applicability of our theory of branching and annihilating random walks to the experimental dataset.

**Signatures of external guidance on the morphology of branched networks.** To go further and test the predicted signatures of external guidance on branch orientations, we next analyzed the distribution of branch angles in the data. As predicted by Eq. (2), we expected the distribution of the angle difference $\psi$ (see Fig. 3a for a schematic) to decay with a variance scaling as $D/\mu f_c$ (see Fig. 3b-d for the distributions $P(\psi)$ obtained from simulations and analytical theory). Comparing theory and experimental data revealed very good agreement, with both single neuron

distributions (see Supplementary Fig. 13 for the individual distributions) and distributions averaged across all data (Fig. 3e) closely following the predicted scaling of the von Mises distribution. Importantly, the single free parameter in this fit (i.e., the variance of the distribution) allows us to estimate $\mu f_c/D$, and thus the relative strength of the global/extrinsic guidance compared with local stochasticity (see Supplementary Note 3 and Supplementary Table 1 for details on the measurements and values of the other parameters, in particular the estimation of the branching probability and branch length). Interestingly, we find intermediate values of $D/\mu f_c \simeq 0.35$, arguing that neuronal morphology is shaped by a combination of both factors.

Such extrinsic guidance provides a simple theoretical mechanism to restrict neuronal growth to a domain characterized by a well-defined opening angle $\bar{\theta}$. Turning to experiments, we found that reconstructed neurons were typically also characterized by such angle, which we estimated as $\langle\bar{\theta}\rangle \simeq 96° \pm 28°$. Theoretically, the average opening angles $\bar{\theta}$ decreased monotonically with increasing field strength $f_c$ (see Fig. 4a for an illustration) with strikingly similar values both in the presence and absence of self-repulsion (see Fig. 4b). Using a simple geometric argument -assuming in particular that this opening angle is determined by the changes in the angle to origin $\theta$ values of the active tips at the boundary of the branching network, we could approximate this angle by:

$$\bar{\theta}_a \simeq 2\chi \, \frac{\log(r_{max})}{f_c} \,, \tag{3}$$

where $r_{max}$ is the radial distance of the furthermost branch from the origin of the network (fixed by the maximal time of network growth). With this approximation, we could fit the numerical data by using a single fit parameter $\chi$ (see Supplementary Note 2 for further details). From the fitted value of the external field $f_c = 0.6$, we predict an opening angle of $\langle\bar{\theta}\rangle \simeq 203° \pm 85°$ (mean ± SD). Although this overestimates the experimental value, we note that this prediction is based on a perfectly radial gradient in 360° without boundary, i.e., assuming neurons can branch backwards. When we confined the theory to a 180° hemispherical region, which seems to reflect the experimental geometries (Supplementary Fig. 11), we obtained average opening angles of $\langle\bar{\theta}\rangle \simeq 110°$, much closer to the data.

Although the existence of an external gradient has not yet been characterized in the zebrafish fin, we note that other features from the comparison between experimental and theoretical data argue in favor of it. For instance, even though self-avoidance can lead locally to aligned branches, these branches grow isotropically in any direction without global cues (see Supplementary Fig. 3 in Supplementary Note 2 for a brief illustration). This is particularly true in low-density regions (which occur stochastically in the simulations) where fewer branches would lead to weaker repulsive cues, and consequently, in the absence of an external gradient, would result in tips deviating from the radial direction. However, examining the data revealed that this did not occur: even in sparse branching regions (e.g., Fig. 2b–d), branches appear as directional toward the fin periphery as in dense branching regions. Furthermore, sparse neurons also showed the same alignment angle distribution as dense neurons in the data (Supplementary Fig. 13c), in contrast to what would happen in the absence of external guidance. This argues for the importance of external guidance to drive robust directional growth independently of stochastic density fluctuations.

**Signatures of self-avoidance on the space-filling properties of branched networks.** Next, we sought a quantitative metric which could distinguish between networks with weak or strong

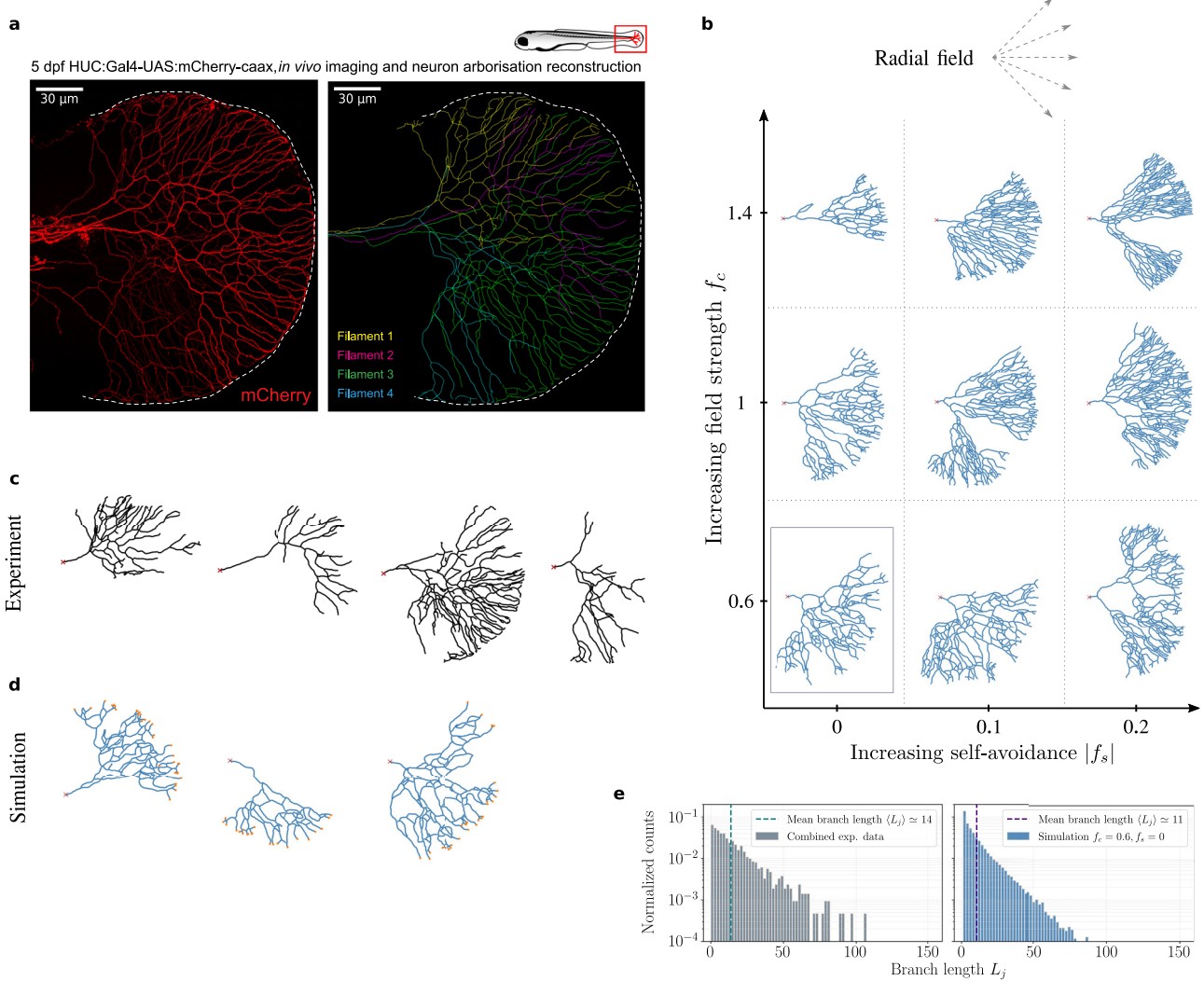

**Fig. 2 Branching and annihilating random walks (BARWs) with radial guidance cue reproduces qualitative features of zebrafish caudal fin innervation.**
**a** Development of the zebrafish nervous system and innervation of the caudal fin (boxed area in the top right cartoon) 5 days postfertilization. (Left)
Confocal image of neuronal cell membranes in the caudal fin imaged via red mCherry fluorescence (HUC:Gal4-UAS:mCherry-caax). Imaged Rohon-Beard
sensory neurons exhibit a clear directionality toward the fin edge (indicated by the dashed white lines). (Right) Different manually reconstructed neuronal
trees color-coded for visualization. **b** Morphology of branched structures with the same model as in Fig. 1, but in a radial external field (dashed arrows, top)
obtained from simulations for different values of the external field strength $f_c$ and self-avoidance $|f_s|$. **c**–**d** Simulations with an intermediate external field
strength ($f_c = 0.6$) and no self-avoidance ($f_s = 0$), corresponding to the boxed region in the morphology diagram (**b**), capture the overall directionality
observed in reconstructed networks (four representative neurons, red cross indicating "origin" of the axon, **c**), but also show some stochasticity in the final
network structure as in the data. Active tips of the simulated branching networks are highlighted in orange. **e** Branch lengths $L_j$ (in normalized units)
obtained from experiments (left) and simulations (right) are distributed exponentially, defining a characteristic length scale $\langle L_j \rangle$ related to the branching
probability $p_b$, as predicted by our theory of stochastic branching. Experimental data are obtained from $n = 8$ reconstructed neuronal filaments from $N = 4$
larvae.

self-repulsion $f_s$ after having estimated $f_c$. Visually, our phase
diagram of neuronal morphology showed that larger self-
avoidance $f_s$ allows for denser networks, as tips can locally avoid
termination and continue growing, compared with branched
networks in the absence of self-avoidance. However, to identify
signatures beyond the coarse-grained metric of overall branch
density, we tested the effect of self-avoidance on the efficiency
of space tiling across length scales[45,46], by quantifying the
fractal dimension $d_f$ of the branching networks (box-counting
method, Fig. 5a) as a function of model parameters. We found
that self-avoidance markedly improved the space-filling prop-
erties of the branching networks (see Fig. 5b, a fractal dimen-
sion close to $d_f = 1$ is expected for very sparse structures, while
a fractal dimension of $d_f = 2$ corresponds to full tiling of space).

Then, we again turned to the experimental data to ask whether
these signatures could be observed. Because the branching rate/
number showed variability across samples, we first explored
this effect, and found a positive correlation between mean
branch probability in a neuron and its fractal dimension
(Supplementary Fig. 14), as expected. Focusing on the four
densest networks to remove this confounding effect, we found
that measuring fractal density in experiments yielded curves
that were consistent with the power-laws predicted by the
simulations (Fig. 5c), with a typical exponent in the range of
$d_f \simeq 1.55 \pm 0.04$ (mean ± SD). This is consistent with our com-
putational screen for relatively small values of self-avoidance
(in the range of $|f_s| = 0 - 0.1$), a feature which was confirmed
by comparing absolute densities between model and data

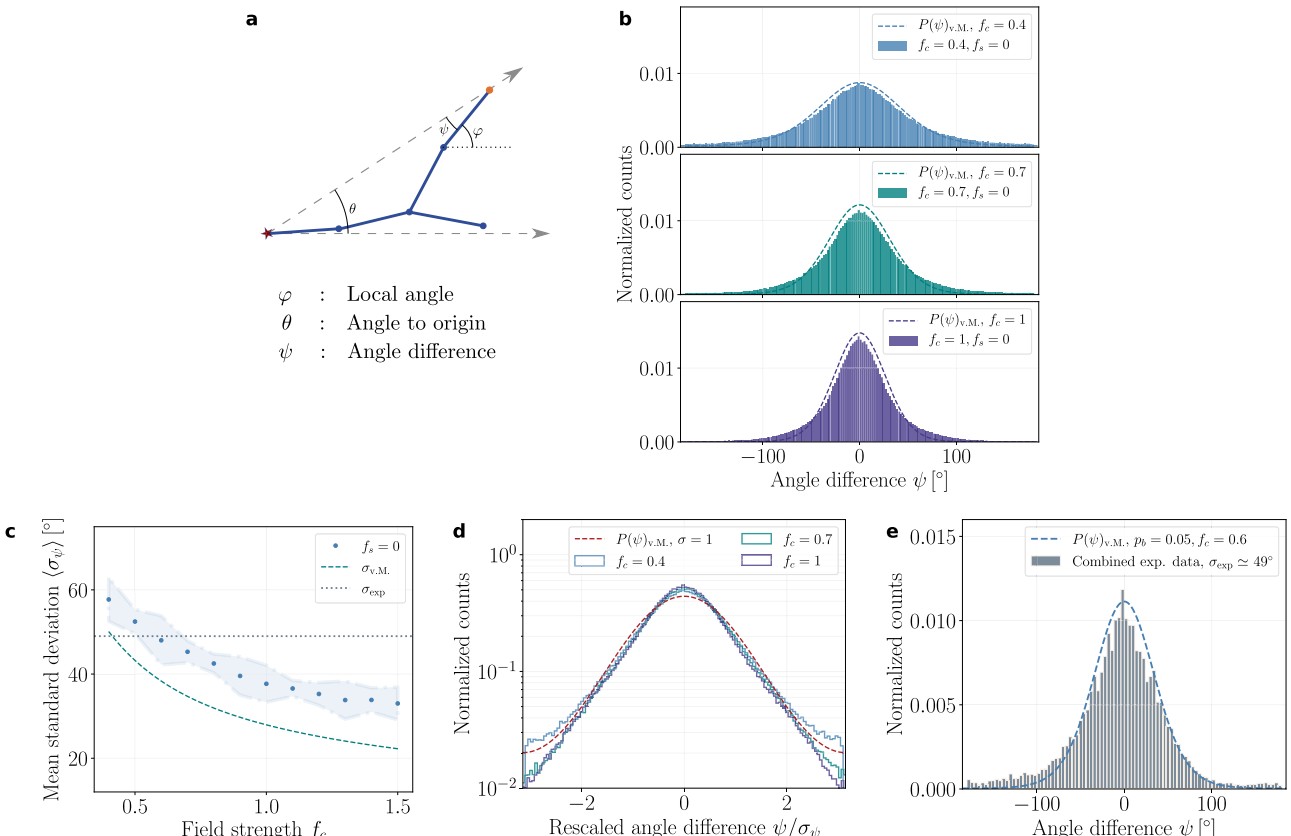

**Fig. 3 Continuum model predicts the alignment of branch segments along the external field both for simulation and experimental data. a** Schematic of branch segments in a radial external field (dashed arrows) highlighting three distinct angles: The *local angle* $\varphi$ of a branch segment with an active tip (orange node), the *angle to origin* $\theta$ (denoted by the star symbol, which determines the extrinsic guidance direction at this point), and the *angle difference* $\psi \equiv \varphi - \theta$ (which tends to be minimized by extrinsic/global guidance). **b** Normalized histograms of the angle difference $\psi$ for different values of $f_c$ and without self-avoidance ($f_s = 0$). The histograms (solid bars) are well-approximated by von Mises distributions (dashed lines) as for the local angle $\varphi$ in a linear gradient (Fig. 1), and as predicted by the continuum model (with $D \simeq 0.04$ and $\mu \simeq 0.2$). **c** Mean standard deviations (SDs) $\langle \sigma_\psi \rangle$ of the angle difference $\psi$ obtained from simulations are close to the mean SD of the experimental data (dotted horizontal line) for an external field strength of $f_c = 0.6$. For comparison, the scaling of SD as a function of $f_c$ predicted by the von Mises distribution is displayed (dashed line). **d** Histograms of angle difference $\psi$ rescaled by their corresponding SDs $\sigma_\psi$ (solid lines) are well-approximated by the von Mises distribution with unit SD (dashed line). **e** Angle difference distribution obtained from the experimental data from $n = 8$ networks (solid bars) compared with the von Mises distribution predicted by the theory for an external field strength of $f_c = 0.6$ (dashed line). The latter value is inferred from the matching of theoretical and experimental values of the SDs displayed in (**c**), and no other fit parameter is used.

(Fig. 5d–f). This argues that although we cannot exclude a small contribution of self-repulsion in locally aligning branches, global external guidance cues play a dominant role in shaping these neuronal structures.

**Effect of dimensionality on the morphology of branched structures**. Finally, although we have so far exclusively considered a two-dimensional (2D) model and simulations, which is a good approximation for the quasi-2D geometry of zebrafish fin innervation, we wished to test whether our findings can be extended to the more general case of branching morphogenesis under external guidance in three-dimensions (3D).

For this generalization, we performed numerical simulations with similar rules for branching, elongation and self-avoidance as before, but allowing tips to evolve in 3D, see Supplementary Note 2 for details. As an exemplary case, we added a constant axial guidance along a single coordinate axis (see Supplementary Movie 2 for an illustration). This could be relevant for the morphogenesis of neurons such as hippocampal granule cells or cortical pyramidal cells[10]. Importantly, a number of quantitative signatures predicted in 2D were unchanged in 3D. For instance,

we found that while a large value for self-avoidance gave rise to locally more aligned and denser branch morphologies (Supplementary Fig. 9a), in strong analogy to the phase diagram in 2D, the overall territory remained minimally influenced by self-avoidance (Supplementary Fig. 9b). Furthermore, the alignment angles (as determined by azimuthal and polar angle coordinates, see Supplementary Fig. 8 for a schematic) both followed the predicted scaling from the analytical theory up to a constant prefactor, with self-avoidance having again negligible influence on these angles (Supplementary Fig. 9c–d). Turning to space-filling efficiency, we found a similar trend where self-avoidance increased the fractal dimension (Supplementary Fig. 9e), although interestingly, the effect was weaker than in 2D, as can be anticipated considering that the probability of branches to interact is much smaller in 3D than in 2D. Overall, this analysis suggests that our key results for 2D branching morphogenesis can be generally extended to 3D situations.

## Discussion
In this work, we have derived an analytical theory, backed by stochastic numerical simulations, of branching morphogenesis

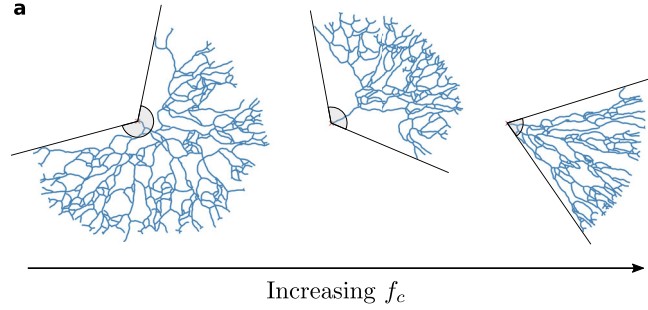

Increasing $f_c$

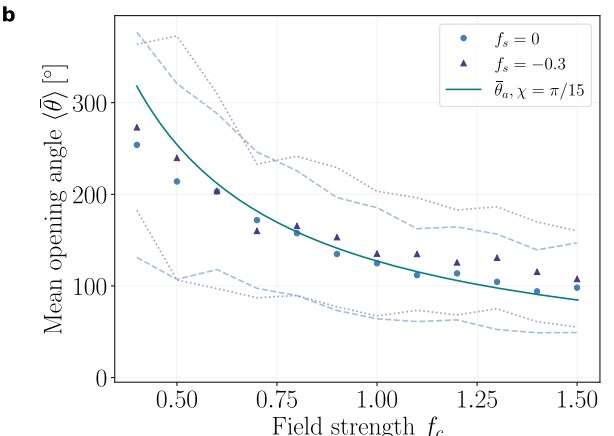

**Fig. 4 The strength of extrinsic guidance determines the territory of the branched structures in simulations, with minimal influence from local self-avoidance. a** Representative simulation snapshots highlighting the changes in the opening angle $\bar{\theta}$ (a proxy for territory size in a radial geometry, gray circular segments) with increasing external field strength $f_c$. **b** Mean opening angles $\langle \bar{\theta} \rangle$ of branched networks decrease monotonically with increasing external field strength $f_c$, and have similar values for networks with zero ($f_s = 0$, circular markers) or with strong ($f_s = -0.3$, triangular markers) self-avoidance. Errors of the averages are determined by the standard deviations and highlighted by the dashed and dotted lines for $f_s = 0$ and $f_s = -0.3$, respectively. The monotonic decrease of $\langle \bar{\theta} \rangle$ obtained from simulations is well-described by the analytical approximation $\bar{\theta}_a$ (solid line), see Eq. (3), with the fit parameter $\chi = \pi/15$.

under both local cues - such as repulsion, branching, and termination - as well as global guidance from external cues. Each of these factors can be tuned to create a variety of complex branched structures. To systematically classify these, and try to understand analytically how each parameter impacts the final structure, we derived a continuum Fokker–Planck theory, which enables us to coarse-grain the parameters of the numerical simulation (branching angles, branching rate, stochasticity in elongation, external guidance strength) into a few relevant coefficients at the macroscopic level. Through this, we have identified a number of generic features in the final branched structures. For instance, a combination of branching/elongation stochasticity in the presence of global guidance cues robustly gives rise to branched structures occupying well-defined spatial domains, but also to a universal scaling law for the alignment of branch angles. This scaling law only depends on the geometry of the problem, i.e., the direction of the external guidance cue both for 2D and 3D territories, with a variance that can be used to extract the relative contribution of external guidance $f_c$ compared with local stochasticity. Self-avoidance of branches, controlled by the parameter $f_s$, on the other hand, has a minimal impact on these features, although it can strongly optimize other morphological parameters such as space-filling properties, as quantified by

overall density or fractal dimension. Interestingly, the branch densities depend strongly on both parameters $f_c$ and $f_s$, which indicates that our predictions on the branch orientations or domain sizes can be used complementarily to disentangle global and local cues.

Our approach here is based on a minimal model to understand the growth of branched structures from simple rules (elongation, branching, guidance, avoidance) within a statistical physics framework. At smaller scales, one would need to take a number of features into account, for instance the specifics of axonal/substrate mechanics during neuronal growth[31,32], to understand what regulates mechanistically each of the parameters that we use in the model. A strength of our "mesoscopic" approach is that it extracts a small number of such coarse-grained parameters, to identify which ones are key at the scale of the overall branching pattern, and thus guiding subsequent, more detailed modelling. Our proposed framework builds upon previous simulations of stochastic branching morphogenesis, which had considered local cues such as branching and repulsion[20,22,33,35]. We find that adding global extrinsic guidance—a key element in different contexts to break the isotropy in tissue growth—in the model gives rise to significantly different dynamics, enriching the phase diagram of possible branching patterns. Furthermore, in addition to the computational/numerical features of this framework, we provide a continuum theory for branching morphogenesis guided by extrinsic cues, which enables us to make simple but generic predictions on testable experimental metrics such as the orientation of branch segments. These predictions are useful to understand branching morphogenesis of organs and neurons, because detailed live-imaging is typically difficult in these systems, so that inferring dynamical information and cues from static snapshots could prove valuable to dwell further into the detailed regulations of these signals. We also note that our strategy of using a mesoscopic theory to infer dynamical growth rules is complementary to other approaches in the field that have addressed the relationship between structure and function on an adult branched organ[15–17]. An interesting next step would be to try to unify the two approaches and test how feedbacks on the growth rules we propose here contribute to design a structure with a desired function.

To begin to test this theory, we have examined the innervation of the zebrafish fin, which proceeds in a simple quasi-2D radial geometry, and, despite the local stochasticity, displays a strong overall radially-oriented bias toward the fin edge. Quantitative reconstructions of several neurons allowed us to test a number of metrics predicted by the theory in the experimental data, such as the distributions of branch lengths and branching angles, or the space-filling properties of individual neurons. In particular, the observation that fin neurons exhibit a clear directional bias with rather well-defined angles can be readily explained in our framework by simply emerging from a global/extrinsic guidance cue which directs single neuronal tips toward the outer edge of the fin. Identifying such an interaction would be a natural next step. It has been shown for instance in the zebrafish pectoral fin that molecules such as BMP or Smoc1 are patterned in a graded way toward the edge during morphogenesis[47], and that innervation of the pectoral fin exhibits a strong variability in sensory neuron morphologies[48]. Overall, a global guidance cue would provide a minimal/complementary explanation to the more involved mechanism of repulsion/tiling between branches of different neurons[49]. Such hetero-avoidance would lead to more refined boundaries between neighboring neurons and could in particular play a role in reducing the domain angles occupied by the individual neurons.

This theoretical framework, although we have applied it here to a specific geometry in neuronal branching, is highly general in

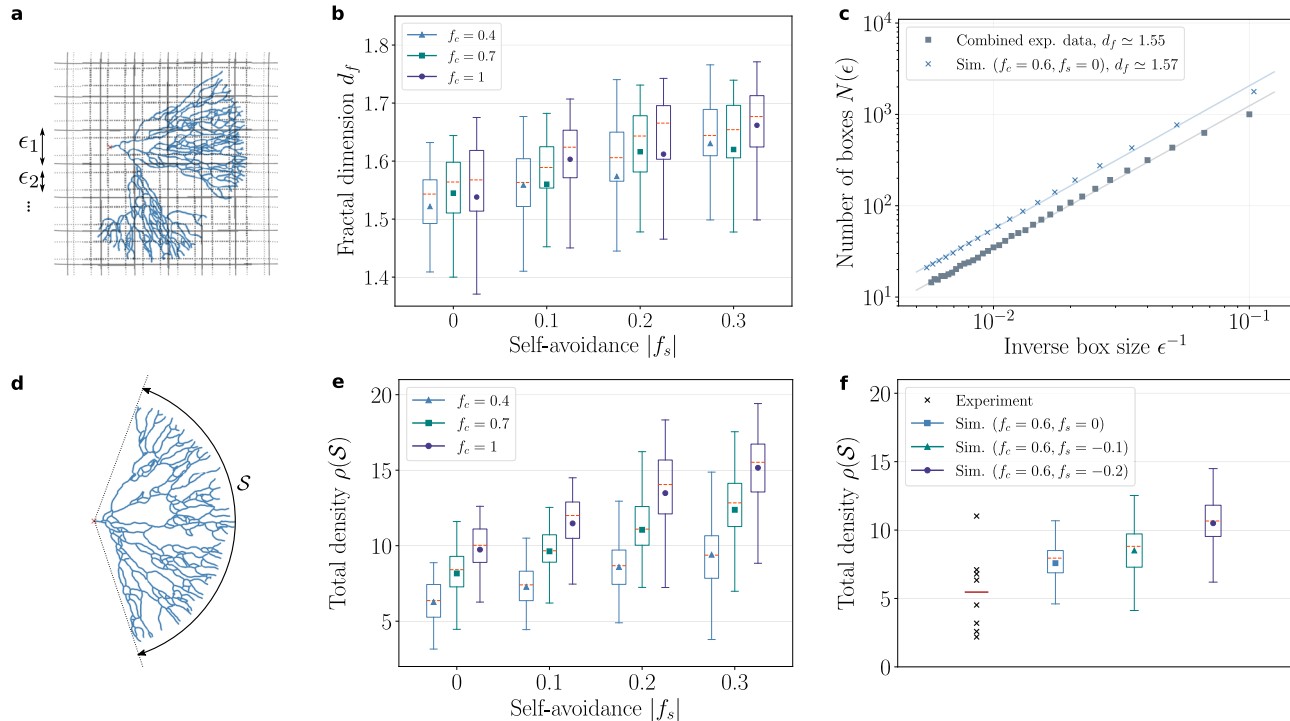

**Fig. 5 Effect of self-avoidance and external guidance on branching density and space-filling properties. a** Fractal dimension of the networks estimated by the box-counting method: Boxes of decreasing sizes $\epsilon$ are used to count the total number of boxes that include at least one skeletonized node. **b** Mean fractal dimensions obtained from the box-counting method increases from $\langle d_f \rangle \simeq 1.52$ to $\langle d_f \rangle \simeq 1.67$ with increasing self-avoidance ($f_s = 0$ to $f_s = -0.3$), whereas large changes in the external field strength (from $f_c = 0.4$ to $f_c = 1$) have a smaller effect on the mean values. **c** Combined experimental data from the densest $n = 4$ networks (circular markers) are consistent with the theoretically predicted power-law: We find a fractal dimension of $d_f \simeq 1.55$, close to the theoretical value $d_f \simeq 1.57$ obtained from the combined data from simulations with $f_c = 0.6$ and $f_s = 0$ (crosses). **d** Average density $\rho(\mathcal{S})$ of a branched network (ratio of the number of branch segments to the arc length $\mathcal{S}$ spanned by the network). **e** Densities $\rho(\mathcal{S})$ of the simulated networks increase markedly both for increasing external field strength $f_c$ and self-repulsion strength $|f_s|$. **f** Densities $\rho(\mathcal{S})$ obtained from experimental data for $n = 8$ filaments (crosses) compared with densities obtained from simulations for $f_c = 0.6$ and $f_s = 0$ (blue box), $f_s = -0.1$ (green box), and $f_s = -0.2$ (purple box). Mean density of the experimental data (red horizontal line) is on the lower end of the densities obtained from simulations even for low repulsion, indicating a small value for the parameter $f_s$. For each parameter choice box plots are obtained from $n = 100$ simulations, with mean and median values denoted respectively by the plot markers and horizontal dashed lines (orange). The boxes are drawn from the first quartile Q1 to the third quartile Q3, and whiskers indicate 1.5 interquartile range (IQR $\equiv$ Q3−Q1), i.e., max = Q3 + 1.5 IQR, min = Q1 − 1.5 IQR.

considering the interplay between external cues and local self-organized rules. In this sense, it could be applied to any branching structure that forms via tip-driven growth, which frequently occurs e.g., during angiogenesis and where similar questions on external guidance vs. local self-organization arise[50,51]. This is strengthened by our findings that the predictions of the model are similar in both 2D and 3D settings. Interestingly, it has recently been proposed that the types of stochastic rules governing tip growth that we model here are conserved for the morphogenesis of various filamentous organisms such as plants or Fungi[52]. Understanding quantitatively the relative contribution of each mechanism is also of key importance for the morphogenesis of branched mammalian organs[53]: Mammary gland, pancreas or late-kidney morphogenesis have been proposed to follow a simpler form of these stochastic models in the absence of external guidance[22], although kidney morphogenesis has been suggested to require stronger self-avoidance (denoted by the parameter $f_s$ in our framework) at early stages to avoid premature termination[54]. This hints at a potentially broad applicability of our framework in a large number of systems, which would be a next step for future research.

## Methods

**Zebrafish transgenic lines and husbandry**. Zebrafish were raised and housed in the Karolinska Institutet core facility following established and approved

procedures. The study was performed in accordance with local guidelines and regulations and approved by Stockholms djurförsöksetiska nämnd. The new transgenic zebrafish strain was generated by injecting UAS:mCherry-caax to Tg(HuC:Gal4; UAS:synaptophysin-GFP) as described below. The resulting F0 transgenic fish express red fluorescent reporter mCherry in cell membranes of a sparse number of neurons, allowing visualization and analysis of neuronal arborization in vivo.

**Cloning**. The expression construct of UAS:mCherry-caax was generated with tol2 kit[55] by recombining p5E-UAS (tol2 kit #327), pME-mCherryCAAX (tol2 kit #550), p3E-polyA (tol2 kit #302), and pDestTol2pA2 (tol2 kit #394). The mRNA of alpha-bungarotoxin was prepared using Addgene plasmid, #69542 as a template and mRNA of pCS2FA-transposase using tol2 kit #396 as a template[56]; in vitro transcription was performed with mMessage mMachine SP6 kit (Thermo Fisher Scientific) and RNA was purified with RNeasy Mini Kit (Qiagen). Zebrafish embryos of Tg(Huc:gal4VP16;UAS:synaptophysin-GFP) were injected with 90 pg of alpha-bungarotoxin mRNA with 10% phenol red and 0.13 M KCl into yolk at one cell stage. Then 20 pg of UAS:mCherry-caax and 20 pg of transposase mRNA were injected with 10% phenol red and 0.13 M KCl into one of the cells at 4–8 cell stage.

**Immunostaining**. For the whole-mount imaging, we anesthetized fish at the 24 hpf, 48 hpf and 5 dpf stages with Tricaine in the same manner as described above, followed by fixation with 4% PFA for 4 h at room temperature. Subsequently, the specimens were permeabilized with three 30 min washes in 100% methanol, washed with PBS supplemented with 0.1% Tween-20 (PBST) five times for 15 min, stained with the primary antibodies in blocking solution (5% normal donkey serum, 10% DMSO, 0.1% Tween-20, in PBS) for 48 h, washed five times in PBST for 30 min, stained with secondary antibodies for 24 h, washed in PBST as described above, and finally dehydrated in 100% methanol with two 30 min washes

and rendered transparent with clearing solution consisting of one part benzyl alcohol and two parts of benzyl benzoate (BABB). The primary antibodies utilized were anti-acetylated tubulin (Neuronal marker, Gene Tex), anti-HuC/HuD neuronal protein and (Abcam), all diluted 1/800 in blocking solution. Alexa fluor 555 donkey anti-rabbit and Alexa fluor 647 donkey anti-mouse (all from Invitrogen) were used as secondary antibodies at a dilution of 1/1000 in blocking solution. Note that tubulin staining was punctiform at high magnification therefore only fish of the mCherry line were used for 3D reconstruction.

**Imaging of live animals**. The expression of mCherry in cell membranes of zebrafish neurons is not uniform, including in the region of the caudal fin, therefore we first screened multiple animals and selected fish which presented mCherry positive signals in the caudal fin. 5dpf fish were anesthetized with tricaine (MS-222, Sigma) in final concentration 200 ug/ml in E3 PTU treated medium. Then five fish samples per dish were immobilized in 500 ul of 0.5% low melting agarose (LMA, Sigma), supplemented with tricaine (200 ug/ml) and placed laterally on glass bottom microwell dish (MatTek, uncoated, 35 mm) using tungsten forceps. After complete polymerization of LMA (40–60 min at room temperature), the droplets containing live fish were covered with Tricaine supplemented with E3-PTU medium to prevent desiccation of the immobilized fish during imaging. Confocal images were acquired using Z-stacks with a Zeiss Zen Blue LSM 800 confocal microscope equipped with Diod lasers 405 nm, 488 nm, 555 nm and 639 nm, Plan Apochromat 10×, Plan Apochromat 20×, and C-Apochromat 40×. Images were processed in Bitplane Imaris 8.0 and exported as .tiff files for further analysis.

**Reconstruction of neuronal filaments**
*Initial (manual) reconstruction of the filaments*. Arborization trees of all visible neurons –the transient Rohon-Beard sensory neurons located in the dorsal spinal cord and innervating the caudal fin integuments[57]—were reconstructed using the pipeline described below. Raw images acquired on live animals were exported from ZEN software to Bitplane Imaris 8.0. For initial reconstruction, Bitplane Imaris tool "Filaments" was used. Resulting images are referred to as "filament trees" in the following. For each image multiple filament trees were acquired. Each tree corresponds to a unique neuron arborization. Each filament tree was then manually analyzed, using native mCherry fluorescence channel as a reference, to eliminate false connections between branches. After manual correction, the tools "smooth filaments" and "center filaments" were applied to further co-localize obtained reconstructions with fluorescence signal. Filaments with confirmed branching pattern and no cross-connectivity artifacts were taken to the next step of analysis (see Supplementary Note 3 for details). All filaments which could not be clearly traced were eliminated from the further analysis. 2D Images of the selected filament trees were exported as separate .tiff files and transferred to ImageJ software. The same set of tools was applied to all filaments: conversion to 8-bit black and white image, skeletonize, analyze skeleton.

*a. Limitations of the experiment:* Prior to the manual reconstruction of the filaments, we performed raw image quality assessment, based on the following parameters. Neurons chosen for reconstruction had a minimal optical overlapping with its neighbors. All raw images were acquired using the same confocal microscopy settings to ensure uniform data resolution. We note that, however, based on the pinhole settings, we could not reliably distinguish between two dots if the Z distance was less than one pinhole, which was set for 1 AU (airy unit) and equal to 1.75 um at 20× objective and to 6.45 at 10× objective. Therefore, any neurites (from two different neuron trees) closer than this distance were discarded from the analysis, creating a small loss in the reconstruction. In a sufficiently dense region of the fin, this loss was evaluated post-hoc to amount to 8.9% of the overall length of all mCherry positive axon arbors (using Imaris tool "filament statistics"). Overall, considering these limitations, a total number of 8 filaments from about 50 fish scanned were qualified for analysis.

**Modelling details**. Detailed descriptions for the derivation of the analytical model and implementation of the BARW simulations, parameter values used in the simulations, and further details on the reconstruction and analysis of the neuronal branches from the experiment can be found in the Supplementary Note. All calculations and results presented in the main text are obtained from $n = 100$ simulations for each parameter choice.

**Reporting summary**. Further information on research design is available in the Nature Research Reporting Summary linked to this article.

## Data availability
Source data obtained from the simulations and experiments that support the findings of this study are provided in a supplementary zip folder, accessible as a DOI link: https://doi.org/10.5281/zenodo.5257160

## Code availability
Custom made Python codes to analyze source data, and an exemplary simulation script to reproduce the main findings are available as Jupyter notebooks in a supplementary zip folder, accessible as a DOI link: https://doi.org/10.5281/zenodo.5257160

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

## Acknowledgements
We thank all members of our respective groups for helpful discussion on the paper. The authors are also grateful to Prof. Abdel El Manira for support and sharing Tg(HUC:-Gal4;UAS:Synaptohysin-GFP), to Haohao Wu for discussion, and thank Elena Zabalueva for the zebrafish schematic. The authors also acknowledge Zebrafish core facility, Genome Engineering Zebrafish and Biomedicum Imaging Core from the Karolinska Institutet for technical support. This work received funding from the ERC under the European Union's Horizon 2020 research and innovation programme (grant agreement No. 851288 to E.H.) and under the Marie Skłodowska-Curie grant agreement No. 754411 (to M.C.U.); Swedish Research Council (to F.L., I.A. and S.H.); Knut and Alice Wallenberg Foundation (F.L. and I.A.); Swedish Brain Foundation (F.L. and S.H.); Ming Wai Lau Foundation (to F.L.); StratRegen (to F.L.); ERC Consolidator grant STEMMING-FROM-NERVE and ERC Synergy Grant KILL-OR-DIFFERENTIATE (to I.A.); Bertil Hallsten Research Foundation (to I.A.); Cancerfonden (to I.A.); the Paradifference Foundation (to I.A.); Austrian Science Fund (to I.A.); and StratNeuro (to S.H.).

## Author contributions
Project initiation: S.H.; Project supervision: S.H. and E.H.; Project conceptualization: S.H., E.H. and M.C.U.; Experiments and reconstructions: D.K. and K.S.; Theoretical model: M.C.U.; Data analysis: D.K., M.C.U. and D.F.; Resources and methodology: F.L. and I.A.; Writing, original draft: E.H. and M.C.U.; Writing, review and editing: S.H., together with inputs from all authors.

## Competing interests
The authors declare no competing interests.
