## [Peer Review File · Nature Communications]

Theory of branching morphogenesis by local interactions and global guidanceReviewers' Comments:

Reviewer #1:

Remarks to the Author:

In brief, the authors provide an analytical theory, backed by simulations, for branching morphogenesis in the presence of external fields. They provide an experimental example on how their theory would apply to zebrafish innervation, but it is my understanding that the main advance of the paper is the analytical, Fokker-Planck like equation the authors derive, and its predictions.

And this is where the problems begin... The work is great, but it is not a scientific manuscript, it is basically an advertisement for the supplement. This was my major grievance with the work. The paper is not self-contained. I understand that there should be some reliance on the supplement, but the statement "Here, we develop a theoretical framework..." is not accurate. It should say "Here, we announce a theoretical framework." The authors offer only a cursory explanation the model, instead referring the reader very heavily to past work to fill in the gaps. Most importantly, they do not say how equation (1), the flagship of the work, comes about. Instead, they spend most of the manuscript comparing the analytics with simulation, and a little bit with experiment. I understand that there are space limitations, but perhaps they authors can condense other parts of the manuscript. For example, I am not sure what is the purpose of the three panels of Fig 2 A. Only one panel is necessary. The rest could have been conveyed in the supplement. Also, in the same figure, panel E is not necessary. It is a visual comparison, without any statistical, quantitative backing. So, to summarize, the authors need to say a bit more about their analytical work (and also, for completeness) their simulations. How are the guidance strengths applied? Exactly how are "decisions" made in branching? All this is not explained concretely with formulas.

Also one minor comment for Fig 1 : it is better if you do not refer to subpanels as "top" and "bottom".

Regarding the "significance" of the work, I personally find it very interesting. The advance compared to past work is the inclusion of the external field term. The comparison between theory and simulation is very convincing. The comparison between simulation and experiment is of course not so strong (space limitations) as there could be alternative explanations for the statistics the authors present. However, I do not find that they authors would need more experimental validation for this work, just the theory part is great. I get the sense that the approach the authors follow might be of interest to a broad audience, but it would be great if the reader could see more of the model/derivation of the analytics in the main paper. Overall, this was great work, and I would have liked to see a clearer presentation of it in the main manuscript.

Reviewer #2:

Remarks to the Author:

This is a well-written and carefully done study of the formation of self-avoiding branched structures. While I find the work interesting and timely, I have some reservations about whether it would be suitable for a broad audience journal. The supporting material of the manuscript is very comprehensive. I am, however, always wary when the SI is longer than the manuscript itself.

Major concerns:

1) The theoretical model is a rather generic model for the formation of branched structures. Such models have been extensively studied in statistical physics for the past 40 years. The novelty is that it is applied to the developmental problem. It is, however, not clear to me what the predictive power of this model is for a developmental system. It is indeed able to capture the shape of branched structures of the innervation of the caudal fin in zebrafish, but it does not tell much about why those structures form in the first place. What I mean here, it is not surprising that a directed branched

structure would arise as a result of a process that involves directed growth and branching. It is, however, not clear what biological mechanism regulates the two parameters in the model, f_s and f_c . For example, it would be very helpful if authors could establish that suppressing or activating appropriate genes, applying mechanical stresses, etc. changes the structure of the fin innervation. This would provide insights into the biological mechanism. Without it, the model is just a rather sophisticated fit to the observed data.

2) While the provided movie indicates that the structures are nearly two-dimensional, I find considering the 2d case only very restrictive. It is known that self-avoidance effects are much stronger in 2d than in 3d. It would be very helpful to compare the effects of the dimensionality of the space on properties of the branched structures.

Minor technical points:

- 1) I would suggest extending the literature to include a bit broader overview of the field.
- 2) I would suggest including key elements of the model into the Methods section in the main article, to make it more self-contained and avoid having to often refer to the SI.
- 3) In what units are f_c and f_s measured?
- 4) What is μ in Eq. (1)?
- 5) It would be helpful to have a clearer explanation of the self-avoidance f_s . What is its functional form?
- 6) In Fig. 1E, making a claim of a power law with less than a decade of data is always questionable. Tick labels in the bottom plot are a bit odd. One might be confused and assume they imply a logarithmic scale.
- 7) Bottom of the left column on pg 2, what is "active growth"?
- 8) Top of the left column on pg 3, what is precisely meant by the phrase "visually more aligned branches"?
- 9) Mid, right column on pg 3, in the phrase "and covering domains of highly different size". How does one quantify "highly different"?
- 10) Fig. 2E, two plots look similar but I am not sure what is the message. Could authors be more specific and quantify the similarity between the two?
- 11) In conclusion, the authors write "This theoretical framework, although we have applied it here to a specific geometry in neuronal branching, is highly general and could be applied to any branching structure such as in angiogenesis..." I find this statement too strong and I am not sure if I agree with it. Blood vessels (in many organisms) form a closed circuit, and more importantly, are organized in hierarchical structures with different diameters. The formation of such networks has been extensively studied, e.g. by Eleni Katifori and her collaborators.

Reviewer #3:

Remarks to the Author:

The authors investigate a theoretical model of branching morphogenesis, which is compared to the zebrafish innervation in experiments. They extended the well-established branching and annihilation random walks (BARWs) model to include external guidance and self-avoidance, and they also derived the coarse-grained Fokker-Planck equation for the probability distribution of the orientation of branches without self-avoidance. The analytical prediction for the steady-state distribution of branch orientations provides new insights regarding the relative importance of external guidance, which was tested in simulations. Furthermore, the authors demonstrated that the statistical properties of branched networks from a theoretical model agree very well with the ones obtained in experiments. This paper would be of interest to a broad spectrum of Nature Communication readers, but it would benefit by addressing the following comments.

1. In the abstract, the authors state that they predict differential signatures of global vs local regulatory mechanisms, but there is a very minimal discussion about it in the main text. Furthermore, it is unclear whether the two effects can be completely disentangled. The opening angle of networks seems to be indeed dominated by the external field strength f_c . The spread of the branch orientation angles is mainly a function of f_c , but it also depends weakly on the strength of self-avoidance f_s as shown in Fig. S5. On the other hand, the fractal dimension and the total density of branches seem to depend both on f_s and f_c . Authors should expand the discussion about the differential signatures of global vs local regulatory mechanisms.
2. The expression for the standard deviation for the von Mises deviation is wrong. Eq. (S26) is actually the circular standard deviation and I am not sure what Eq. (S27) is representing. In the absence of external guiding ($\nu=0$) the stationary probability distribution is uniform with the standard deviation $\sigma=\pi/\sqrt{3}$, but the expression in Eq. (S27) would suggest that the standard deviation is diverging.
3. Explain how the simulation is done when both the external field guidance and self-avoidance are included. Are both effects included simultaneously to displace the active tip before rescaling the segment length back to ℓ ? If this was not the case and these effects were done sequentially, then the magnitude of self-avoidance parameter f_s would be irrelevant because $r^*-r = -f_s \cdot p_s$ gets rescaled such that $|r^*-r|=\ell$
4. Is the self-avoidance in Eq. (S29) only affected by the points on neighboring branches or are the active tips also repelled by other inactive segments on the same branch?
5. Explain what is the source of stochasticity for the model of external field via tip displacement in section S2.2.1. Are orientation angles perturbed during the branch elongation before the reorientation due to external field? If simulations were completely deterministic, then the results in Fig. S6 don't make sense.
6. Explain what geometric arguments were used to derive the opening angles in Eq. (3) and related Eq. (S30). I think the authors are implying that the arguments are similar to the ones that were used to derive Eq. (S4). However, it is also unclear how that equation was derived. One can use the law of sines to derive Eq. (S3), but I don't see a straightforward way to derive Eq. (S4). This should be clarified.
7. When discussing the exponential distribution of branch lengths in Fig. 2F on page 3, it would be useful to mention that the characteristic length scale is related to the branching probability parameter p_b , which is estimated in the Supplementary Information.
8. The sentence below Eq. (1) states that the steady-state solution is largely independent of the form of the external field. It is unclear what is meant by that because authors considered only one specific form of the external field in this Equation.
9. Explain what is the meaning of the symbols on the box-and-whisker plots in Fig. 5B,E. Are symbols indicating the mean values?
10. At the beginning of the results section, it should be explicitly stated that the 2D networks will be investigated because some of the examples in the introduction also refer to 3D networks.
11. Report the numerical values of D and μ either in the main text or in the figure captions. In principle, readers can extract their values by using Eq. (S18), but then they need to search for the values of other model parameters that are scattered throughout the Supplementary Information. It would help the readers to summarize typical values of simulation parameters in a single table.

12. Explain how is the averaging done for the mean standard deviation in Figs. 1E, 3C, S4C, S5C, S6C, and opening angles in Figs. 4B, S5E. How many simulations were included to perform these averages?

13. The fitting parameter ϕ_b for the opening angle in Eq. (3) may be confused with the angle associated with branching. It would be better to use a different symbol for the fitting parameter.

14. What was the rationale for choosing the maximal jump sizes $\phi_e = \pi/10$ and $\phi_b = \pi/2$ for angle values?

15. In Eq. (S22) there should be no time on the left-hand side. It would also be good to use the subscript P^{st} since this is the differential equation for the stationary distribution.

16. Provide the volume and page numbers for references [16] and [35]. Provide the bioRxiv number for reference [30].

RESPONSE TO REFEREE COMMENTS - NCOMMS-21-16563

August 24, 2021

This document describes our detailed point-by-point responses to the specific comments of the three referees. Each comment Cx.x) is quoted in *italics* followed by our response Rx.x). At the end of our responses, we append a marked copy of the revised main text (MT-marked) and the supplementary information (SI-marked), with the changes since the first submission highlighted in blue.

Detailed responses to Reviewer 1

In brief, the authors provide an analytical theory, backed by simulations, for branching morphogenesis in the presence of external fields. They provide an experimental example on how their theory would apply to zebrafish innervation, but it is my understanding that the main advance of the paper is the analytical, Fokker-Planck like equation the authors derive, and its predictions.

We thank Reviewer 1 for their critical reading of the manuscript and for the detailed comments and suggestions.

C1.1) *And this is where the problems begin... The work is great, but it is not a scientific manuscript, it is basically an advertisement for the supplement. This was my major grievance with the work. The paper is not self-contained. I understand that there should be some reliance on the supplement, but the statement "Here, we develop a theoretical framework..." is not accurate. It should say "Here, we announce a theoretical framework." The authors offer only a cursory explanation the model, instead referring the reader very heavily to past work to fill in the gaps. Most importantly, they do not say how equation (1), the flagship of the work, comes about. Instead, they spend most of the manuscript comparing the analytics with simulation, and a little bit with experiment.*

R1.1) We agree with the reviewer that because most of the modelling details were included in the supplemental note (originally to keep the paper accessible to non-theory audiences), key aspects of our theoretical framework remained rather inaccessible upon first reading of the main text. To streamline the paper better, we have now divided it into sub-sections, and in particular now added a new sub-section “Derivation of the continuum model” in the beginning of the Results section (page 2 in MT-marked), where we describe in detail the key steps and assumptions used to derive Eq.(1). We have also provided further details on the simulations in the section “Influence of self-avoidance, stochasticity and external guidance on the morphology of branched networks” (beginning of page 2 in MT-marked).

C1.2) *I understand that there are space limitations, but perhaps they authors can condense other parts of the manuscript. For example, I am not sure what is the purpose of the three panels of Fig 2 A. Only one panel is necessary. The rest could have been conveyed in the supplement.*

R1.2) The two (leftmost and middle) panels of Fig.2A showed that most filaments were successfully reconstructed manually by comparing these with the raw images of the neuronal membranes. However, we agree with the reviewer that this information does not need to be included in the main text as a separate figure, and therefore now moved the middle panel of Fig.2A to the supplement, see Fig.S10 in SI-marked.

C1.3) *Also, in the same figure, panel E is not necessary. It is a visual comparison, without any statistical, quantitative backing.*

R1.3) We agree with the referee, and have now moved this to Supplementary Fig.S12, again to streamline the manuscript better. We also agree that the previous version was too qualitative in showing that the topology of subtree sizes exhibits a strong heterogeneity (both in the simulation and experimental data). To clarify this, we have now added a quantification of the cumulative subtree size distributions of both data and simulations (see the new panel E in Fig.S12). We have described this in a new text piece in the

main text (page 5 in MT-marked, last paragraph before the subsection “Signatures of external guidance on the morphology of branched networks”).

C1.4) *So, to summarize, the authors need to say a bit more about their analytical work (and also, for completeness) their simulations. How are the guidance strengths applied? Exactly how are "decisions" made in branching? All this is not explained concretely with formulas.*

R1.4) As mentioned above, in addition to the new subsection “Derivation of the continuum model” in the main text, we have now included a more detailed description of the rules used in the simulation: In particular, we explained how the external field strength is applied by the transition probabilities of alignment angles ($A(\psi)$ and $B(\psi)$) as

$$A(\psi) - B(\psi) = -f_c \sin(\psi), \quad \text{with} \quad A(\psi) + B(\psi) = 1 \quad (\text{R1})$$

that reweight the “forward” and “backward” elongation probabilities of a branch segment as $p_e A(\psi)$ and $p_e B(\psi)$, respectively. On the other hand, we have added a more detailed description of the self-avoidance as well as branching rules used in the simulations, see the corresponding new text pieces in the main text, see page 2 in MT-marked (penultimate paragraph before subsection “Derivation of the continuum model”).

C1.5) *Also one minor comment for Fig 1 : it is better if you do not refer to subpanels as "top" and "bottom".*

R1.5) We now labelled the individual elements in Fig.1 with different letters and do not refer to them as “top” and “bottom”.

C1.6) *Regarding the "significance" of the work, I personally find it very interesting. The advance compared to past work is the inclusion of the external field term. The comparison between theory and simulation is very convincing. The comparison between simulation and experiment is of course not so strong (space limitations) as there could be alternative explanations for the statistics the authors present. However, I do not find that they authors would need more experimental validation for this work, just the theory part is great. I get the sense that the approach the authors follow might be of interest to a broad audience, but it would be great if the reader could see more of the model/derivation of the analytics in the main paper. Overall, this was great work, and I would have liked to see a clearer presentation of it in the main manuscript.*

R1.6) We thank the reviewer for their appreciation of our theoretical framework and experimental validation, as well as their constructive suggestions. We hope that with the modifications made, particularly regarding the modelling details and the figures, the presentation in the main text has become more transparent and comprehensible.

Detailed responses to Reviewer 2

This is a well-written and carefully done study of the formation of self-avoiding branched structures. While I find the work interesting and timely, I have some reservations about whether it would be suitable for a broad audience journal. The supporting material of the manuscript is very comprehensive. I am, however, always wary when the SI is longer than the manuscript itself.

We thank Reviewer 2 for their detailed comments and suggestions. We agree that the presentation of the manuscript, as well as the ratio of information between SI and main text was sub-optimal, as also underlined by the other reviewers. We have thus re-written, expanded and streamlined the main text to make it more self-contained (adding a section on model derivation and moving the experimental methods in main text, as well as expanded the introduction and discussion with more broad references on the fields – see below for details) .

Major concerns:

C2.1) *The theoretical model is a rather generic model for the formation of branched structures. Such models have been extensively studied in statistical physics for the past 40 years. The novelty is that it is applied to the developmental problem. It is, however, not clear to me what the predictive power of this model is for a developmental system. It is indeed able to capture the shape of branched structures of the innervation of the caudal fin in zebrafish, but it does not tell much about why those structures form in the first place. What I mean here, it is not surprising that a directed branched structure would arise as a result of a process that involves directed growth and branching. It is, however, not clear what biological mechanism regulates the two parameters in the model, f_s and f_c . For example, it would be very helpful if authors could establish that suppressing or activating appropriate genes, applying mechanical stresses, etc. changes the structure of the fin innervation. This would provide insights into the biological mechanism. Without it, the model is just a rather sophisticated fit to the observed data.*

R2.1) We agree with the referee that adding external guidance cue is indeed qualitatively expected to give rise to directed branched growth. However, and as the referee states, how such a global cue quantitatively translates into the structure of developing organs at multiple scales remain unclear. Indeed, although a large body of literature on branched organs has been devoted to explain the design principles of adult networks such as fractality and optimal transport, the mechanisms via which these networks actually form remains poorly understood. In this regard, our minimal model for self-organizing branching structures provides a distinct explanatory framework compared to previous theoretical studies, which mostly implemented the geometry (Cuntz et al., PLoS Comp. Biol. 2010) or intrinsic rules of the branching process (Palavalli et al., Curr. Biol. 2021) strongly based on the specific experimental phenomenology.

Furthermore, we believe that the type of mesoscopic rules that we propose here could be useful in the future, precisely because they provide a bridge between large scale observables (structure of branched networks) and the underlying molecular/biological mechanisms that encode them. In particular, self-repulsion (parameter f_s) has been shown to be regulated by a number of possible molecular mechanisms, such as diffusible cues or contact-mediated adhesion codes (Villar-Cerviño et al., Neuron 2013). External guidance (parameter f_c) has similarly been shown to be achievable either via mechanical (Koser et al., Nat. Neurosci. 2016) or biochemical gradients (Lanoue and Cooper, Dev. Biol. 2019). Importantly, our analytical Fokker-Plank derivation underlines the fact that the resulting collective behavior of many tips/branches is largely independent of the exact microscopic details regulating f_c and f_s . We realize that the Discussion of the previous version of the manuscript was not very informative in this sense, and tried to expand/clarify this in this revised version (new text pieces on page 9 in MT-marked). Concerning the predictive power of the model, we believe that this generality of the analytical theory still leads to several nontrivial predictions such as the scaling of branch orientations or territory as a function of external field strength, which we do not believe were reported previously, and were testable in our dataset. Thus, the model can be used both qualitatively – to test for the presence of features such as external cues during stochastic branching morphogenesis, but also more quantitatively to infer the presence and relative strength of local interactions shaping the tissue during development.

Concerning the underlying biological mechanism, we agree with the reviewer that perturbation experiments would provide further tests of the model. Unfortunately, the molecular cues that regulate directionality in zebrafish fin neuronal morphogenesis remain unknown, and there could thus be a multitude of potentially redundant mechanisms operating in this rather unexplored system (for instance, external cues such as gradients of signaling molecules, e.g. guidance molecules such as netrin, or morphogens, or of mechanical stiffness). Thus, we think this question should be addressed in a series of subsequent experimental studies.

C2.2) *While the provided movie indicates that the structures are nearly two-dimensional, I find considering the 2d case only very restrictive. It is known that self-avoidance effects are much stronger in 2d than in 3d. It would be very helpful to compare the effects of the dimensionality of the space on properties of the branched structures.*

R2.2) We thank the reviewer for this interesting comment. To explore the effect of dimensionality on the branching patterns, we have now extended our model with 3D branching and annihilating random walk simulations in an axial field (uniform field along the x-axis), see the new subsection “Effect of dimensionality on the morphology of branched structures” in the Results of the main text, as well as section S2.6 “Simulation of BARWs with external guidance in three-dimensions” in SI-marked. Importantly, we now show that a number of signatures predicted in 2D are unchanged in 3D. For instance, we find that while a large value for self-avoidance increases the density of the final network, the overall territory remains minimally influenced by self-avoidance, see new Fig.S9B, see also new added Video for a movie of a typical 3D simulation. Furthermore, the alignment angles (as determined by azimuthal and polar angle coordinates) both follow the predicted scaling from the analytical theory up to a constant prefactor, and self-avoidance has almost no influence on the alignment angles (see Fig.S9C-D, in strong analogy to our results in 2D). Turning to space-filling efficiency, we also find, as surmised by the reviewer, that self-avoidance has a smaller influence on the final branch pattern in 3D: the fractal dimension of branched networks depended rather weakly on self-avoidance, showing only a marginal increase with increasing self-avoidance, see Fig.S9E.

Minor technical points:

C2.3) *I would suggest extending the literature to include a bit broader overview of the field.*

R2.3) We have now included a more comprehensive discussion of the literature, see new text pieces in pages 1 (first paragraph), 2 (left column) and 9 (Discussion) in MT-marked. We also added 9 new references in Introduction and Discussion, to compare and contrast our results to other approaches on branched organs, in particular on ideas around optimality and function of specific branching patterns.

C2.4) *I would suggest including key elements of the model into the Methods section in the main article, to make it more self-contained and avoid having to often refer to the SI.*

R2.4) We thank the referee for this suggestion. As alluded to above, to make the main text more self-contained, we have now added the new subsection “Derivation of the continuum model” in the beginning of the Results section, where we explain the key steps and assumptions of the model. We have also added new text pieces in the main text to more transparently explain the rules and assumptions used in the simulation, see new text pieces in page 2 of MT-marked in the subsection “Influence of self-avoidance, stochasticity and external guidance on the morphology of branched networks”. We have also moved all of the experimental methods and details on the reconstruction analysis to a “Methods” section in main text, as suggested by the referee (page 10-11 of the main text).

C2.5) *In what units are f_c and f_s measured?*

R2.5) These two parameters are dimensionless: The field strength f_c is defined (i) in the tip displacement-based simulations as the prefactor for the displacement $f_c \mathbf{p}_c$, where \mathbf{p}_c is the polarity vector aligned with the field orientation. (ii) In the implementation of the external field via biased probabilities, f_c is defined by the transition probabilities $A(\psi)$ and $B(\psi)$ as $A(\psi) - B(\psi) = -f_c \sin(\psi)$. The self-avoidance parameter f_s , on the other hand, is the prefactor for the displacement $f_s \mathbf{p}_s$, where \mathbf{p}_s is the self-interaction vector determined by the local density of neighboring branches, see Eq.(S29). We have now clarified these in

the revised text pieces in subsections S2.2.1 and S2.2.2, as well as with the new Fig.S5 in the SI-marked.

C2.6) *What is μ in Eq. (1)?*

R2.6) μ can be regarded as a motility/advection coefficient, which has the explicit functional form given in Eq.(S18). We have now added the functional form in the main text, see page 3 (left column) in MT-marked (subsection “Derivation of the continuum model”).

C2.7) *It would be helpful to have a clearer explanation of the self-avoidance f_s . What is its functional form?*

R2.7) f_s is a prefactor that scales the net vector of self-avoidance given in Eq.(S29). We now briefly refer to its functional form in the main text, see text piece in page 2 (right column, top) of MT-marked (subsection “Influence of self-avoidance, stochasticity and external guidance on the morphology of branched networks”).

C2.8) *In Fig. 1E, making a claim of a power law with less than a decade of data is always questionable. Tick labels in the bottom plot are a bit odd. One might be confused and assume they imply a logarithmic scale.*

R2.8) We agree in general with the reviewer on claims of power laws, but remark that Fig.1E is a plot for 12 different choices of the parameter f_c covering a wide range of network topologies. The bottom plot is indeed a log-log plot of the data shown in the top panel. The power law behavior is interesting for us because it is a prediction of the analytical model. However, we have softened the claim on the power-law nature of the curve, only saying that it is “consistent with a power-law relation” predicted by the analytics (caption of Fig.1, now labelled as panel F).

C2.9) *Bottom of the left column on pg 2, what is “active growth”?*

R2.9) We have now removed the term “active” from the corresponding sentence.

C2.10) *Top of the left column on pg 3, what is precisely meant by the phrase “visually more aligned branches”.*

R2.10) We have now tried to be more clear by replacing the phrase “visually more aligned” with “denser branches with increased local alignment”.

C2.11) *Mid, right column on pg 3, in the phrase “and covering domains of highly different size”. How does one quantify “highly different”?*

R2.11) We agree with the reviewer that this statement was not precise enough. One metric to quantify domain size in 2D is given by the opening angle $\bar{\theta}$ of the networks. Because we discuss opening angles separately in the subsection “Signatures of external guidance on the morphology of branched networks”, we have now removed this text piece.

C2.12) *Fig. 2E, two plots look similar but I am not sure what is the message. Could authors be more specific and quantify the similarity between the two?*

R2.12) These two plots highlight the overall heterogeneity in the subtree structures of the branched networks. We have now moved this panel to Fig.S12 as we agreed that it was not strictly necessary for a main figure. Furthermore, to provide a quantitative metric, we have also added a plot of the cumulative subtree size distributions from experiments and simulations which showed close correspondence, see the new panel E in Fig.S12, and the corresponding new text piece in the main text, see page 5 in MT-marked (last paragraph before the subsection “Signatures of external guidance on the morphology of branched networks”).

C2.13) *In conclusion, the authors write “This theoretical framework, although we have applied it here to a specific geometry in neuronal branching, is highly general and could be applied to any branching structure such as in angiogenesis...” I find this statement too strong and I am not sure if I agree with it. Blood vesicles (in many organisms) form a closed circuit, and more importantly, are organized in hierarchical structures with different diameters. The formation of such networks has been extensively studied, e.g. by Eleni Katifori and her collaborators.*

R2.13) We agree with the reviewer that the initial version of this sentence was too strong – we meant that this could apply quite generically to instances of tip-driven branching morphogenesis. Although a number of networks such as blood vessels are hierarchical and indeed form via different processes (e.g. remodeling of a connected plexus), angiogenesis can also proceed via similar tip-driven events, especially during the early stages of development. To be more accurate about the limits and applicability of our framework, we have now changed the corresponding text piece, see page 9 in MT-marked (last paragraph). We note that we have also added several papers by Katifori and her collaborators in the Introduction and Discussion of our revised manuscript, as suggested by the referee, which help to give a broader view of the field indeed.

Detailed responses to Reviewer 3

The authors investigate a theoretical model of branching morphogenesis, which is compared to the zebrafish innervation in experiments. They extended the well-established branching and annihilation random walks (BARWs) model to include external guidance and self-avoidance, and they also derived the coarse-grained Fokker-Planck equation for the probability distribution of the orientation of branches without self-avoidance. The analytical prediction for the steady-state distribution of branch orientations provides new insights regarding the relative importance of external guidance, which was tested in simulations. Furthermore, the authors demonstrated that the statistical properties of branched networks from a theoretical model agree very well with the ones obtained in experiments. This paper would be of interest to a broad spectrum of Nature Communication readers, but it would benefit by addressing the following comments.

We thank Reviewer 3 for their critical reading of the manuscript and for the detailed comments and suggestions.

C3.1) *In the abstract, the authors state that they predict differential signatures of global vs local regulatory mechanisms, but there is a very minimal discussion about it in the main text. Furthermore, it is unclear whether the two effects can be completely disentangled. The opening angle of networks seems to be indeed dominated by the external field strength f_c . The spread of the branch orientation angles is mainly a function of f_c , but it also depends weakly on the strength of self-avoidance f_s as shown in Fig. S5. On the other hand, the fractal dimension and the total density of branches seem to depend both on f_s and f_c . Authors should expand the discussion about the differential signatures of global vs local regulatory mechanisms.*

R3.1) We agree that our discussion in general did not summarize clearly enough the different effects of each metric. The referee is correct in that for some metrics explored, such as the branch density and fractal dimensions, the effects of f_c vs. f_s are not easily distinguished, whereas others such as angle distributions can be used to fit one of the parameter f_c efficiently. However, from a theoretical perspective, this means that we can use the latter to first fit the parameter f_c with little ambiguity, and then use this constraint to fit the other parameter f_s on other metrics. From a biological perspective, this could imply complementary roles for each mechanism to shape branched structures with different properties and/or functions. We have now elaborated carefully on this issue in the discussion of each of these metrics, see revised/new text piece on page 9 in MT-marked (first paragraph in Discussion), and we have also summarized the signatures at the end of each corresponding sections of the main results (page 5, left column for external guidance, page 7, right column for self-avoidance.)

C3.2) *The expression for the standard deviation for the von Mises deviation is wrong. Eq. (S26) is actually the circular standard deviation and I am not sure what Eq. (S27) is representing. In the absence of external guiding ($\nu=0$) the stationary probability distribution is uniform with the standard deviation $\sigma=\pi/\sqrt{3}$, but the expression in Eq. (S27) would suggest that the standard deviation is diverging.*

R3.2) We thank the referee for this important technical comment. The expression for the **circular variance** (Eq.(S26)) is valid for von Mises distribution, where $\mathcal{R}(\nu)$ corresponds to the mean resultant length. The expression for the circular standard deviation, see Eq.(S27), as taken from the book “Mardia and Jupp, Directional Statistics” (John Wiley & Sons, 2009), see Eq. 2.3.11 therein, however, does indeed lead to a divergence when applied to von Mises distribution for $\nu \rightarrow 0$. An alternative way is to use the “linear” version for the SD by $\sqrt{1 - \mathcal{R}(\nu)}$, which would converge for $\nu \rightarrow 0$ but will in general underestimate the circular SDs. We note that Eq.(S27) therefore provides indeed a good approximation for the SD of the von Mises distribution for sufficiently large ν , but have now clarified this issue in the corresponding new text pieces below Eq.(S27) in SI-marked.

C3.3) *Explain how the simulation is done when both the external field guidance and self-avoidance are included. Are both effects included simultaneously to displace the active tip before rescaling the segment length back to ℓ ? If this was not the case and these effects were done sequentially, then the magnitude of self-avoidance parameter f_s would be irrelevant because $r^*-r = -f_s$. p_s gets rescaled such that*

$$|r^*-r| = \ell$$

R3.3) We thank the referee for prompting this important clarification. We note that **only for the tip displacement-based implementation** of external guidance, i.e. for the results shown in subsection S2.5 and Fig.S7, the effects of f_s and f_c are performed sequentially in the simulation setup. For all simulations included in the main text, the external field modifies the elongation probabilities and does not deterministically change the position of the active tip. Furthermore, even for the sequential application of f_c and f_s , the rescaling does not imply that the magnitude of the self-avoidance f_s parameter is irrelevant: The displacement of the active tip away from neighboring branch segments at timepoint t is controlled by f_s as $\mathbf{r}^*(t) - \mathbf{r}(t) = -f_s \mathbf{p}_s$. However, the rescaling is done to conserve the step size with respect to the previous time point, such that $\mathbf{r}^*(t) - \mathbf{r}(t - \tau) = \ell$. Therefore the rescaling only influences the length of the segment connecting the displaced node and the previous node, where f_s will determine the position of the displaced node and therefore will lead to different positions as its magnitude changes. We now explained in more detail how this rule is implemented in the simulation, see extended text in subsection S2.2.2, and the detailed schematic in the new panel A in Fig.S5 in SI-marked.

C3.4) *Is the self-avoidance in Eq. (S29) only affected by the points on neighboring branches or are the active tips also repelled by other inactive segments on the same branch?*

R3.4) Active tips are only repelled by branch segments on the neighboring branches, i.e. they cannot be repelled by segments in their own branch (but can annihilate still on self-contact). Note that this choice has little influence for networks that do not exhibit a strong rotational diffusion of their branches (i.e. branches are highly unlikely to annihilate on “themselves”), see extended text below Eq.(S29) in SI-marked.

C3.5) *Explain what is the source of stochasticity for the model of external field via tip displacement in section S2.2.1. Are orientation angles perturbed during the branch elongation before the reorientation due to external field? If simulations were completely deterministic, then the results in Fig. S6 don't make sense.*

R3.5) The referee is correct in that, during an elongation event, the active tip of a branch first undergoes a small rotational diffusion regardless of the external field strength. The external field then subsequently acts on the position of the tip to reorient it by a factor determined by f_c . We have now emphasized this point in the new text piece in subsection 2.2.1 in SI-marked.

C3.6) *Explain what geometric arguments were used to derive the opening angles in Eq. (3) and related Eq. (S30). I think the authors are implying that the arguments are similar to the ones that were used to derive Eq. (S4). However, it is also unclear how that equation was derived. One can use the law of sines to derive Eq. (S3), but I don't see a straightforward way to derive Eq. (S4). This should be clarified.*

R3.6) We have now revised the subsection S2.3 to include a more detailed derivation of the approximation for the opening angles, and included a schematic for visualization, see new panel B in Fig.S5. We have also clarified the derivation of Eq.(S4), for which one needs to use the sine rule two times for two different triangles, see new text piece below Eq.(S4) in SI-marked and improved panel B of Fig.S1.

C3.7) *When discussing the exponential distribution of branch lengths in Fig. 2F on page 3, it would be useful to mention that the characteristic length scale is related to the branching probability parameter p_b , which is estimated in the Supplementary Information.*

R3.7) The characteristic length scale (average branch length) is indeed determined by the inverse branching probability. However, due to frequent annihilation events, inverse branch length in fact overestimates the branching probability, which is why we estimated the latter by the ratio of branch segments to the total size of branch generations until a terminal branch is reached. We now added a new text piece to mention this relation in Fig.2 caption.

C3.8) *The sentence below Eq. (1) states that the steady-state solution is largely independent of the form of the external field. It is unclear what is meant by that because authors considered only one specific form of the external field in this Equation.*

R3.8) We thank the referee for this comment. The steady-state solution in fact applies both to radial and axial fields that we had explored in the main text. The only modification is given by the definition of the alignment angle: In an axial field, the alignment angle is simply given by the local angle φ , whereas in a radial field the angle difference ψ takes the role of the alignment angle. In fact, we now show that the azimuthal and polar angles of the branch segments of a 3D network can also be defined as alignment angles in an axial field and follow the same steady-state profile, see the new subsection S2.6 in SI-marked. To emphasize this more transparently, we have revised the corresponding text piece, see page 3 in MT-marked (first paragraph in subsection “Comparison between analytical model and simulations”).

C3.9) *Explain what is the meaning of the symbols on the box-and-whisker plots in Fig. 5B,E. Are symbols indicating the mean values?*

R3.9) We apologize: the sentence explaining the meaning of these symbols was placed wrongly in the figure caption. We have now moved the corresponding sentence to the end of the caption of Fig.5.

C3.10) *At the beginning of the results section, it should be explicitly stated that the 2D networks will be investigated because some of the examples in the introduction also refer to 3D networks.*

R3.10) We have now extended our simulations to explore stochastic branched structures under external guidance in three-dimensions, see also point C2.2 above. We added the new subsection “Effect of dimensionality on the morphology of branched structures” to summarize our basic results on three-dimensional networks in the main text, while providing a detailed description for the general 3D case in the new subsection S2.6, as well as with the new Figs.S8 & S9 in the SI-marked.

C3.11) *Report the numerical values of D and μ either in the main text or in the figure captions. In principle, readers can extract their values by using Eq. (S18), but then they need to search for the values of other model parameters that are scattered throughout the Supplementary Information. It would help the readers to summarize typical values of simulation parameters in a single table.*

R3.11) We have now added a table in the supplementary information, see Table S1 in SI-marked, to list the parameters used in the simulation. We furthermore refer to the values of D and μ in the corresponding figure captions, see revised captions of Figs.1 & 3.

C3.12) *Explain how is the averaging done for the mean standard deviation in Figs. 1E, 3C, S4C, S5C, S6C, and opening angles in Figs. 4B, S5E. How many simulations were included to perform these averages?*

R3.12) All simulation results in the main text were obtained from $n = 100$ simulations for each parameter choice, see the text piece in the new subsection “Modelling details” in Methods of the main text. For the supplementary figures different numbers of simulations were performed for each parameter choice, which we have now added at the end of the corresponding figure captions in the SI (see Figs. S4, S6, S7 & S9).

C3.13) *The fitting parameter ϕ_b for the opening angle in Eq. (3) may be confused with the angle associated with branching. It would be better to use a different symbol for the fitting parameter.*

R3.13) We thank the referee for this suggestion, and we have now changed the symbol of the fitting parameter from φ_b to χ .

C3.14) *What was the rationale for choosing the maximal jump sizes $\phi_e = \pi/10$ and $\phi_b = \pi/2$ for angle values?*

R3.14) We had two reasons to constrain the angular jump sizes during branching to be uniformly distributed in $[\pi/10, \pi/2]$: (i) Practical reasons for the simulations implied that if the angle between two newly born branches was smaller than $\pi/5$ we observed frequent annihilation events for these new progenies. Therefore we constrained the minimal jump after a branching event to be $\pi/10$. This then provided a natural choice for the maximal jump for an elongation event, without the need to introduce a further parameter. In principle, however, different values for these maximal jumps can be used to explore the effects of these local noise factors. (ii) Empirically when we looked at the branch angle distributions from the neuronal filaments, we saw that at least 85% of the angles between two newly born progenies were distributed within the range $[\pi/5, \pi]$, which furthermore justified our choice, see Fig.R1 below. An

Figure R1: Distributions of branching angles (angle between the two newly born progenies) for the experimental data from $n = 8$ individual filaments. Shaded region (gray) indicates the angle range $[\pi/5, \pi]$ which was used to constrain the branching angles in the BARW simulation setup.

alternative way would be using a fit to the experimental data, but due to the rather strong fluctuations in these $n = 8$ individual distributions, we preferred using uniform distributions (principle of indifference) for these jump probabilities. We have now added a text piece in subsection S2.1 in SI to elaborate this point.

C3.15) *In Eq. (S22) there should be no time on the left-hand side. It would also be good to use the subscript $P^{\wedge}st$ since this is the differential equation for the stationary distribution.*

R3.15) We thank the reviewer for this careful observation. We have now changed the symbols as suggested, see the updated Eq.(S22).

C3.16) *Provide the volume and page numbers for references [16] and [35]. Provide the bioRxiv number for reference [30].*

R3.16) We have now corrected the corresponding references (now labelled [28], [45] and [51]).

Reviewers' Comments:

Reviewer #3:

Remarks to the Author:

Authors have addressed all of my previous concerns, and I recommend publication.

Minor comments:

* the stationary probability distribution on the right hand side of Eq. (22) should have the 'st' subscript

* In the captions of Fig. 1, the sentence 'Schematic of the model and resulting branching morphologies.' refers to panels (A-C) and not (A-B).

We thank the Referee 3 for their careful reading and recommendation of our manuscript for publication. In the following, the final comments of the Referee 3 are quoted in italics, followed by our responses.

Minor comments:

** the stationary probability distribution on the right hand side of Eq. (22) should have the 'st' subscript*

Response: We have now added the superscript 'st' to the symbol 'P' of the stationary probability distribution.

** In the captions of Fig. 1, the sentence 'Schematic of the model and resulting branching morphologies.' refers to panels (A-C) and not (A-B).*

Response: We have now corrected this.